# Compressive Failure Characteristics of 3D Four-Directional Braided Composites with Prefabricated Holes

**DOI:** 10.3390/ma17153821

**Published:** 2024-08-02

**Authors:** Xin Wang, Hanhua Li, Yuxuan Zhang, Yue Guan, Shi Yan, Junjun Zhai

**Affiliations:** 1Department of Engineering Mechanics, Harbin University of Science and Technology, Harbin 150080, China; 2Department of Engineering Mechanics, Beijing Institute of Astronautical Systems Engineering, Beijing 100076, China; 3College of Aeronautics and Astronautics, North China Institute of Aerospace Engineering, Langfang 065000, China

**Keywords:** 3D4d braided composites, prefabricated holes, damage tolerance, failure mechanism

## Abstract

The low delamination tendency and high damage tolerance of three-dimensional (3D) braided composites highlight their significant potential in handling defects. To enhance the engineering potential of three-dimensional four-directional (3D4d) braided composites and assess the failure mode of hole defects, this study introduces a series of 3D4d braided composites with prefabricated holes, studying their compressive properties and failure mechanisms through experimental and finite element methods. Digital image correlation (DIC) was used to monitor the compressive strain on the surface of materials. Scanning acoustic microscope (SAM) and scanning electron microscopy (SEM) were used to characterize the longitudinal compression failure mode inside the material. A macroscopic model is established, and the porous materials are predicted by using the general braided composite material prediction theory. While reducing the forecast cost, the error is also controlled within 21%. The analysis of failure mechanisms elucidates the damage extension mode, and the porous damage tolerance ability aligns closely with the bearing mode of braided material structure. Different braiding angles will lead to different bearing modes of materials. Under longitudinal compression, the average strength loss of 15° specimens is 38.21%, and that of 30° specimens is 8.1%. The larger the braided angle, the stronger the porous damage tolerance. Different types of prefabricated holes will also affect their mechanical properties and damage tolerance.

## 1. Introduction

Three-dimensional (3D) braided materials are garnering significant interest across industries due to their high shear strength, enhanced interlaminar fracture toughness, low delamination, high damage tolerance, and capability to produce net intricate shapes through numerical control [1,2,3,4,5,6,7,8]. Currently, 3D braided composites are being actively utilized in aerospace, maritime, rail transit, and sports sectors [9,10,11]. In principle, the low delamination tendency and high damage tolerance of 3D braided composites can effectively reduce the influence of hole defects, and perforation can increase its assembly possibility. However, at present, the research on the hole defects in braided composites is not mature enough. To enhance the engineering potential of 3D braided composites and investigate the effects of hole defects, a series of 3D braided composites with prefabricated holes were developed and their compressive properties and failure modes systematically analyzed through experiments and the finite element method (FEM).

3D braided composites exhibit excellent compression and load-bearing capabilities due to their interlaced yarn braiding structure. In engineering applications, notably in structural design, material compressibility is of paramount importance. Currently, researchers have conducted comprehensive studies on the compression performance of 3D-FDBCs from three perspectives: experimental, FEM, and theoretical. Studies indicate that the compression behavior of 3D braided composites is significantly influenced by the braiding angle, fiber volume fraction, and loading direction. Fang et al. [12] employed a Representative Volume Element (RVE) in conjunction with damage theory and the finite element method to examine the uniaxial compressive properties of braided composites across varying braiding angles. The calculation model accounts for fiber dislocation and the longitudinal shear nonlinearity of the braided yarn. Numerical findings reveal that the compressive behavior of braided composites with low braiding angles is highly sensitive to initial fiber defects in the braided yarns. The strength of braided composites across various braiding angles is determined by distinct micro-failure modes. Zhu et al. [13] developed a parametric FEM for three-dimensional four-directional (3D4d) braided composites. The findings reveal that both the braiding angle and fiber volume fraction significantly impact damage evolution and compressive strength. Gao et al. [14] explored the strength prediction, failure modes, and stress and strain fields in 3D braided composites with void defects using FEM. They noted that an increase in void defects led to a slight decrease in the strength and modulus of the composites. Ge et al. [15] introduced a two-scale progressive damage approach incorporating an RVE. The findings indicate that pore defects diminish the matrix properties, significantly affecting the strength and damage mechanisms of three-dimensional braided composites with a large braiding angle. Ai et al. [16] examined the longitudinal and out-of-plane compressive properties, along with the progressive damage, of 3D5d braided composites under various parameters. As the fiber volume fraction increases, both the longitudinal and non-planar mechanical properties of the fibers improve. Increasing the braiding angle results in a decrease in the longitudinal mechanical properties, out-of-plane strength, and modulus of the composites. Du et al. [17] took into account the interaction between yarns in developing their 3D5d parametric FEM. They discovered that variations in braiding angle and fiber volume fraction altered the primary failure mode and initial damage location in the composites. Liu et al. [18] explored the effects of manufacturing defects and braiding angle on the impact compression performance of 3D carbon fiber/epoxy resin circular braided composite pipes through numerical simulation. The findings indicate that the braiding angle influences how fiber bundle and resin defects affect the compressive strength, initial modulus, and failure strain of the braided pipes.

The damage tolerance of 3D braided materials has prompted researchers to investigate their response to openings. To understand the impact of holes on 3D braided materials and predict hole strength, researchers have explored how holes affect the tensile and compressive properties of these materials. Gause and Alper [19] conducted an extensive study on the structural properties of 3D braided composites. Experiments on braided specimens with 6.35 mm (1/4 in.) diameter open holes show no reduction in tensile strength due to the hole. According to the tensile and compressive tests, Liang et al. [20] examined the macroscopic mechanical properties and failure behaviors of 3D braided composites with a central circular hole, assessing strength variations across different hole radii. The results show that 3D braided composites predominantly fracture transversely along the notch, exhibiting greater notch insensitivity compared to laminated composites. In addition to experiments, the strength prediction of woven materials with holes began as early as the 1970s. Whitney and Nuismer [21] investigated the strength characteristics of fiber-reinforced composites with circular holes, introducing the Point Stress Criterion (PSC) and Average Stress Criterion (ASC) based on stress distribution. These criteria aim to predict the uniaxial tensile strength of laminated composites with variations in shape and thickness. Liang et al. [22] applied the PSC to estimate the open-hole strength of 3D braided composites. The findings indicate that, compared to traditional braided laminates, 3D braided composites retain a higher proportional residual strength, even with varying sizes of circular holes. Gao et al. [23] used the PSC, ASC, and progressive damage analysis under various failure criteria to individually assess the tensile strength of braided composite plates with holes. Hwan et al. [24] enhanced the prediction of strength for all the kinds of braided composite plates with central holes using refined PSC and ASC methods. Relative to experiments, numerical methods offer a more efficient and cost-effective way to uncover the failure mechanisms of 3D braided composites [25,26,27,28,29]. However, on the basis of a limited number of experiments, the strength prediction of braided materials with holes requires additional research. Moreover, the majority of these single-hole predictions rely on the PSC and ASC, methods originally devised for laminated composites containing through the thickness discontinuities. This reliance introduces significant limitations and compromises that merit attention.

Prior studies indicate that 3D braided materials exhibit superior damage tolerance compared to laminates, with a single hole causing limited loss in mechanical properties. Consequently, double-hole is introduced to extend the assembly application of braided composites. Additionally, analyzing the failure mechanisms of 3D braided composites with perforation defects enhances the prediction of defective material properties and supports engineering applications with valuable data. This paper proposes six different types of prefabricated holes (double holes) in 3D4d braided composites and examines their impact on the compressive mechanical properties and failure behavior of specimens with two braiding angles (15° and 30°). Longitudinal quasi-static compression tests were conducted on specimens with prefabricated holes. Material damage was detected using digital image correlation (DIC), a scanning acoustic microscope (SAM), and scanning electron microscopy (SEM). Simultaneously, based on the four-step braiding method and Zhu’s RVE [13], a parametric model of a large-size 3D4d braided composite plate was developed. The FEM used in paper, recognized for its applicability to 3D braided composites with complete structure, predicts their macroscopic properties by simplifying hole defects into structural defects, thus lowering the prediction costs for such composites. The FEM of the compression process validates the experimental results and elucidates the materials’ failure behavior.

## 2. Experiment

The compression experiment specimens, made of 3D4d braided composites, utilize Toray 12K T700 carbon fiber for the yarn and TDE-86 epoxy resin for the matrix. The experimental materials were produced by Hubei Philly and prepared by China Quartz Glass Co., LTD. (Jingzhou, China). The specimens feature braiding angles of 15° and 30°. Manufacturing process variations result in different volume fractions across all samples. Knuckle dimensions vary with the knitting angle. Refer to Table 1 for material parameters. Figure 1a displays the composite compression fixture. The compression experimental setup is illustrated in Figure 1b. Quasi-static compression experiments yielded macroscopic longitudinal compressive stress-strain curves and fracture behavior at a compression rate of 0.5 mm/min. DIC features monitor the specimen in real-time during compression, capturing deformation details such as displacement and strain by analyzing speckle pattern changes on the specimen’s surface pre- and post-deformation. Following compression failure, internal damage was assessed using ultrasonic microscopic imaging and electron microscopy. Experiments were conducted at room temperature.

There are 6 kinds of prefabricated holes in the specimen. Two holes, each 6 mm in diameter, were created at the specimen’s center, spaced 12 mm and 24 mm apart in the transverse, longitudinal, and yarn braiding directions, labeled as type 1, 2, and 3 holes, respectively. A 6 mm diameter twist drill was used as the drilling tool. Figure 1c,d illustrates that while holes drilled with a twist drill have more burrs compared to those made with a double-edged keyway milling cutter, these burrs minimally affect the specimen. Although holes created by the double-edged keyway milling cutter are cleaner, this method may roll up and potentially damage some of the specimen’s fiber bundles. Figure 1e displays the six types of prefabricated holes.

## 3. Finite Element Analysis

### 3.1. Mesoscopic Scale Parametric Model and Mesh

The formation of 3D4d braided composite fiber preforms involves a four-step braiding method, illustrated in Figure 2a. Rectangular yarn carriers operate under specific guidelines to weave yarns into preforms. The x-y plane can record the position of the yarns at each step. Figure 2b displays the surface morphology of the braided material, where the braiding angle *α* is the angle between the yarns and the longitudinal axis. Figure 2c shows the plane position of the yarn for a 4 × 33 rectangular braiding captured in one cycle of the four-step braiding method. The color curve represents the trajectory of selected yarns throughout a complete cycle. The black arrow is the moving direction of the row or column. The h denotes the distance between steps in the four-step braiding method along the z axis, simulating the braided material’s pitch length. The w represents the spacing between yarns on the x and y axes, mimicking the pitch width of the braided material. In order to ensure the high fiber volume ratio of the model and describe the internal structure of the material efficiently and correctly, the model is established based on ZHU’s parameterized RVE [13]. Based on Zhu’s parameterized RVE, the spatial relationship of yarns within the braided material can be described as follows:(1)$$w=\frac{1}{2}\sqrt{\frac{\pi \sqrt{1+2{tan}^{2}\alpha }}{V_{f}}}D_{y}$$
(2)$$h=\frac{1}{4}\sqrt{\frac{1+2{tan}^{2}\alpha }{V_{f}{tan}^{2}\alpha }}D_{y}=\frac{w}{4tan\alpha }$$
(3)$$D_{y}=\sqrt{\frac{4\lambda }{\pi \rho }}$$

*D_y_* represents the equivalent diameter, *λ* the linear density, *ρ* the density, *α* the braiding angle, and *V_f_* the fiber volume fraction.

By projecting the yarn’s position on the *x*-*y* plane onto the matrix and implementing the braiding algorithm in Python, we can track each yarn’s precise location on the *x*-*y* plane at every step, resulting in a segmented line representation of the yarn’s trajectory. The manufacturing process of braided composites is regarded as a nonlinear solid mechanics problem, which can accurately analyze the yarn interaction and cross-section deformation neglected in the traditional topological model. Sun proved that the yarn in the 3D braided preform is curved, which is contrary to the linear assumption of the topological model [30]. Furthermore, the segmented line does not maintain the continuity of the fiber bundle. Therefore, employing an interpolation algorithm to generate an appropriate trajectory curve is essential. This study employs B-spline curves to simulate fiber trajectory curves. Compared with the Bézier curve, the B-spline curve offers the advantage of segmentation. Modifying any control vertex alters only the adjacent curve segment’s shape, not the overall curve trend, thereby effectively addressing local control and connectivity issues [31,32]. Furthermore, the third-order B-spline curve can well adapt to the internal structure of 3D4d braided composites. The k-order B-spline curve, featuring *n* + 1 control points, and its basis functions *B_i,k_*(*u*), as delineated by the De Boer-Cox recurrence formula, are defined as follows:(4)$$p\left(u\right)=\left[\begin{array}{cc} \begin{array}{cc} P_{0} & P_{1} \end{array} & \begin{array}{cc} \cdots & P_{n} \end{array} \end{array}\right]\left[\begin{array}{c} \begin{array}{c} B_{0,k}\\ B_{1,k} \end{array}\\ \begin{array}{c} \cdots \\ B_{n,k} \end{array} \end{array}\right]=\sum_{i=0}^{n}P_{i}B_{i,k}\left(u\right)$$
(5)$$B_{i,k}=\left\{\begin{array}{c} \left\{\begin{array}{c} 1{,u}_{i}{\leq u<u}_{i+1}\\ 0,other \end{array},\right.k=1\\ \frac{{u-u}_{i}}{u_{i+k-1}-u}B_{i,k-1}\left(u\right)+\frac{u_{i+k}-u}{u_{i+k}{-u}_{i+1}}B_{i+1,k-1}\left(u\right),k\geq 2 \end{array}\right.$$

The spatial structure of yarns can be defined by four specific parameters: *λ*, *ρ*, *α*, and *V_f_*. Assuming a sufficiently high fiber volume fraction and no interference between yarns, the geometric model parameters are detailed as Table 2.

Figure 3a,b depict the fiber structure in the parametric model of 3D braided materials. Achieving uniform size for yarn structures with two braiding angles without cutting is challenging. Therefore, the diameter and spacing of the model’s holes are made equivalent to the experiment’s prefabricated holes, based on model sizes. The assemblies and meshes of the model, including the embedded constraints for the yarn and matrix, are illustrated in Figure 3c. Employing embedding constraints significantly reduces computational effort. Concurrently, it allows for segmenting the yarn and matrix into C3D8R high-quality meshes, maintaining a high fiber volume fraction. Figure 3d displays the mesh configuration around the holes. Mesh quality is not compromised by structural issues in yarns that remain uncut at the opening.

### 3.2. Progressive Damage Model

Current strength prediction for hole defects in 3D4d braided composites requires additional research, with most studies concentrating on single hole defects in limited-area plates. Without addressing the defect’s unique mechanical behaviors, the simulation focuses solely on the macro-structural issues resulting from defects, using established failure criteria and damage evolution models for composite materials. This universal FEM serves as a reference for defect research in other braided materials and can be extended to materials with multiple hole defects in principle. This study employs the 3D Hashin failure criterion to predict initial failure, which is defined as follows [33,34]:

Failure mode of fiber longitudinal tension
(6)$$\xi_{Lt}={\left(\frac{\sigma_{11}}{X_{t}}\right)}^{2}+\frac{\sigma_{12}^{2}}{S_{12}^{2}}+\frac{\sigma_{31}^{2}}{S_{13}^{2}}-1\geq 0$$

Failure mode of fiber longitudinal compression
(7)$$\xi_{Lc}={\left(\frac{\sigma_{11}}{X_{c}}\right)}^{2}-1\geq 0$$

Failure mode of fiber transverse tension
(8)$$\xi_{Tt}=\frac{1}{Y_{t}^{2}}{\left(\sigma_{22}+\sigma_{33}\right)}^{2}+\frac{1}{S_{23}^{2}}\left(\sigma_{23}^{2}-\sigma_{22}\sigma_{33}\right)+\frac{1}{S_{12}^{2}}\left(\sigma_{12}^{2}+\sigma_{31}^{2}\right)-1\geq 0$$

Failure mode of fiber transverse compression
(9)$$\xi_{Tc}=\frac{1}{Y_{c}}\left[{\left(\frac{Y_{c}}{2S_{23}}\right)}^{2}-1\right]\left(\sigma_{22}+\sigma_{33}\right)+\frac{1}{4S_{23}^{2}}{\left(\sigma_{22}+\sigma_{33}\right)}^{2}+\;\frac{1}{S_{23}^{2}}\left(\sigma_{23}^{2}-\sigma_{22}\sigma_{33}\right)+\frac{1}{S_{12}^{2}}\left(\sigma_{12}^{2}+\sigma_{31}^{2}\right)-1\geq 0$$

Failure mode of matrix tension
(10)$$\xi_{Mt}={\left(\frac{\sigma_{11}}{F_{t}}\right)}^{2}-1\geq 0$$

Failure mode of matrix compression
(11)$$\xi_{Mc}={\left(\frac{\sigma_{11}}{F_{c}}\right)}^{2}-1\geq 0$$

*X_t_* represents the tensile strength along the yarn direction, while *X_c_* denotes the corresponding compressive strength. *Y_t_* and *Y_c_* refer to the tensile and compressive strengths, respectively, in the transverse direction. *S*_12_, *S*_23_, and *S*_13_ indicate shear strengths. *F_t_* and *F_c_* describe the matrix’s tensile and compressive strengths, corresponding to different damage modes. Upon meeting the initial damage criteria, the subsequent damage evolution is as follows:(12)$$\left\{\begin{array}{c} d_{I}=\frac{\varepsilon_{I}^{e,f}}{\varepsilon_{I}^{e,f}-\varepsilon_{I}^{e,i}}\left(1-\frac{\varepsilon_{I}^{e,i}}{\varepsilon_{I}^{e}}\right)\begin{array}{cc} & \left(d_{I}\in \left(0,1\right),I=L_{t}{,L}_{c},T_{t},T_{c},M_{t},M_{c}\right) \end{array}\\ \varepsilon_{I}^{e,f}=\frac{2G_{I}}{\sigma_{I}^{e,i}l} \end{array}\right.$$

*d_I_* is defined as the damage variable. $\varepsilon_{I}^{e,i}$ denotes the elastic strain upon initial damage, while $\varepsilon_{I}^{e,f}$ represents the ultimate failure elastic strain when the damage variable equals one. *G_I_* signifies the fracture energy associated with the failure mode, and *l* is the characteristic length of an element, derived from the cube root of its volume. The damage modes for both yarn and matrix are categorized into three primary damage variables along the *L*, *T*, and *M* directions, as follows [34]:(13)$$\left\{\begin{array}{c} D_{L}=max\left(d_{Lt},d_{Lc}\right)\\ D_{T}=max\left(d_{Tt},d_{Tc}\right)\\ D_{M}=D_{L}=D_{T} \end{array}\right.$$

The stiffness degradation matrix is defined as follows [34]:(14)$$C_{d}=\left[\begin{array}{cc} \begin{array}{ccc} b_{L}^{2}C_{11} & b_{L}b_{T}C_{12} & b_{L}b_{T}C_{13}\\ & b_{T}^{2}C_{22} & b_{T}^{2}C_{23}\\ & & b_{T}^{2}C_{33} \end{array} & \begin{array}{ccc} & 0 & \end{array}\\ \begin{array}{ccc} & sym & \end{array} & \begin{array}{ccc} b_{LT}C_{44} & & \\ & b_{LT}C_{55} & \\ & & b_{TT}C_{66} \end{array} \end{array}\right]$$
(15)$$b_{L}=1-D_{L},b_{T}=1-D_{T},b_{LT}={\left(\frac{2\left(1-D_{L}\right)\left(1-D_{T}\right)}{2-D_{L}-D_{T}}\right)}^{2},b_{TT}={\left(1-D_{T}\right)}^{2}$$

### 3.3. Defining Material Properties

In 3D4d braided composites, carbon fiber (T700/12K) serves as the reinforcing phase, while epoxy resin (TDE-86) constitutes the matrix. Here, the yarn is classified as a transversely isotropic material, and the matrix as an isotropic material. Detailed composition parameters of components are provided in Table 3 and Table 4 [12]. The Chamis empirical formula is used to effectively predict the performance parameters of materials, and their effective elastic properties and strength are calculated as follows [35,36]. Compared with various forms of PSC combined with ASC to directly predict the local stress at a single hole, the Chamis empirical formula is more suitable for this paper to study the macro-mechanical properties of 3D braided composites with holes.
(16)$$\left\{\begin{array}{c} \begin{array}{c} X_{t}=\varphi \eta_{t}S_{ft}\\ X_{c}=\varphi \eta_{c}S_{fc}\\ Y_{t}=\left[1-\left(\sqrt{\varphi }-\varphi \right)\left(1-\frac{E_{m}}{E_{f22}}\right)\right]S_{mt} \end{array}\\ \begin{array}{c} Y_{c}=\left[1-\left(\sqrt{\varphi }-\varphi \right)\left(1-\frac{E_{m}}{E_{f22}}\right)\right]S_{mc}\\ S_{12}=S_{13}=\left[1-\left(\sqrt{\varphi }-\varphi \right)\left(1-\frac{G_{m}}{G_{f12}}\right)\right]S_{ms}\\ S_{23}=\left[\frac{1-\sqrt{\varphi }\left(1-\frac{G_{m}}{G_{f12}}\right)}{1-\varphi \left(1-\frac{G_{m}}{G_{f12}}\right)}\right]S_{ms} \end{array} \end{array}\right.$$

*φ* denotes the yarn filling coefficient. To derive the yarn’s directional tensile (*X_t_*) and compressive (*X_c_*) strengths, the yarn’s inherent tensile (*S_ft_*) and compressive (*S_fc_*) strengths are adjusted using the correction factor *η*. The correction factor *η* is related to the braided structure. Strengths perpendicular to the yarn direction—tensile (*Y_t_*), compressive (*Y_c_*), and shear (*S*_12_, *S*_13_, *S*_23_)—are determined by the matrix’s tensile (*S_mt_*), compressive (*S_mc_*), and shear (*S_ms_*) strengths. The matrix’s elastic (*E_m_*) and shear (*G_m_*) moduli, along with the fibers’ elastic modulus (*E_f_*_22_), longitudinal shear modulus (*G_f_*_12_), and transverse shear modulus (*G_f_*_23_) perpendicular to the yarn direction, are specified.

## 4. Results and Discussion

### 4.1. Damage Extension and Failure Mechanism

The intricate internal structure of 3D4d braided composites limits the effectiveness of 2D DIC images processed by the VIC-2D system. Thus, analysis is confined to the principal strain distribution on the specimen surface at 100% compressive peak strength, as depicted in Figure 4. Additionally, Figure 4 illustrates the specimen surfaces at two braiding angles at the onset of failure and the immediate subsequent damage. Upon failure initiation in the 15° specimen, longitudinal cracks develop at both the upper and lower ends of the prefabricated holes, with damage subsequently spreading laterally to create additional surface longitudinal cracks. The DIC principal strain map reveals that significant strain primarily localizes at the upper and lower ends of the prefabricated holes, exhibiting a flame-like shape. Contour lines are densest at the periphery of the high strain zones, indicating a steep gradient in change. Furthermore, all regions of the DIC image show positive values, suggesting that the entire plate bulges outward under compression in the 15° specimen. The initial damage in the 15° specimen is identified as surface tensile damage resulting from the specimen’s convex deformation under pressure.

Upon the onset of failure in the 30° specimen, minor independent damage emerges near the prefabricated hole, with subsequent lateral extension leading to more extensive damage resembling fish scales. The DIC principal strain map for the 30° specimen reveals a complex surface strain pattern, with positive and negative strain areas intermingling. The initial damage in the 30° specimen manifests within the composite, influencing its macroscopic deformation.

Figure 5 illustrates the failure modes of specimens with two different braiding angles. The 15° specimens exhibit brittle fracture upon compressive failure, characterized by fiber folds and rifts. Fiber folds result in surface cracks crossing the prefabricated holes. Scanning reveals a clear internal fracture line, diverging from the crack pattern on the specimen’s surface. This discrepancy arises because the rift in braided composites is oblique, leading to shear failure in the main yarn. The 30° specimens show a fish-scale damage pattern upon compressive failure. Failure manifests as discontinuous, localized stratification over short distances. For the 30° specimens, the primary load-bearing mechanism involves interactions both among yarns and between yarn and matrix. Shear failure in certain material regions results from the progressive damage following local delamination.

Figure 6 displays the electron micrograph of the material fracture. Figure 6a reveals the flat fracture surface of the carbon fiber in the 15° specimen. Brittle failure causes the yarn to abruptly sever across a large area. Figure 6b shows the disordered fracture surface of carbon fiber in the 30° specimens. This disordered pattern arises because local delamination, following initial failure, induces varied mechanical behaviors in the yarn before it breaks.

The configuration of prefabricated holes significantly impacts the compression behavior of 3D braided composites. As shown in Figure 7, Type 1 prefabricated holes (arranged horizontally) exhibit a high coupling in mechanical behavior, with failure spreading laterally across both holes. This is primarily due to the transverse direction of compression damage propagation. Type 2 prefabricated holes (arranged longitudinally) show low mechanical coupling, with damage predominantly extending through just one of the holes. However, for specimen 15-2-12-(1), the prefabricated holes exhibit a coupled relationship. This coupling is attributed to the large deformation failure in the 15° specimen, causing a longitudinal tensile crack that connects both prefabricated holes. The reduced coupling results not just from the transverse spread of compression damage but also from the 3D braided structure that alters the straightforward vertical transmission of force. The mechanical coupling of type 3 prefabricated holes falls between that of types 1 and 2. Specifically, specimen 30-3-24-(1) demonstrates no coupling between its prefabricated holes.

### 4.2. FEM’s Results

The experiment shows the material failure under the influence of comprehensive factors, and the influence of the pre-drilled hole material is further verified by finite element simulation. Figure 8 illustrates that the FEM failure outcomes for the 15° specimens closely align with experimental results, although simulating initial tensile cracks poses a challenge for this model. The embedding constraint synchronizes the displacement of the yarn and matrix, preventing the FEM from displaying the local delamination observed in the 30° specimens. Without embedded constraints, simulating the overall model with a high volume ratio becomes challenging. Consequently, cell deletion represents the complete failure of the 30° specimens. The damage distribution map obtained by simulation shows that the fiber bundle, serving as the primary load-bearing structure, initiates damage at the prefabricated hole. L damage is longitudinal compression damage, and T damage is transverse compression damage. As damage spreads laterally, various prefabricated holes contribute to differing extents of damage propagation.

Figure 9 illustrates that the damage around the two Type 1 prefabricated holes progressively increases from initial to complete damage. The two cavities are involved in absorbing compression energy throughout the process. Compression damage to the matrix occurs almost simultaneously with fiber damage. However, the extent of matrix damage is relatively limited, confined to areas of fiber damage. For the 12mm prefabricated hole, transverse damage expands progressively, eventually encompassing the entire hole. Longitudinal compression damage dictates the direction of damage propagation. For the 24 mm prefabricated hole, damage does not significantly extend until complete failure occurs. Destruction occurs more abruptly and quickly. This indicates that the 24 mm Type 1 prefabricated hole has a greater energy absorption capacity than the 12 mm Type 1 hole.

Figure 10 demonstrates that, without interference from other factors, only one of the Type 2 prefabricated holes is involved in energy absorption and failure throughout the process. Following initial damage, the other prefabricated hole remains uninvolved in the failure process. Matrix damage occurs in sync with fiber damage, within a limited area. For the 12 mm prefabricated holes, damage extends over a relatively brief period. Until complete destruction, compression damage remains largely confined to a small area around the prefabricated hole. The damage propagation for the 24 mm prefabricated hole is similar to that of the Type 1 12 mm hole. The energy absorption capacity of the Type 2 12 mm prefabricated hole exceeds that of the Type 2 24 mm hole.

Figure 11 reveals that the failure pattern of the Type 3 prefabricated hole mirrors that of the Type 1 hole at 12 mm, and aligns with the Type 2 hole at 24 mm. The damage distribution pattern derived from the finite element method corresponds well with the observed experimental failure behavior.

### 4.3. Analysis of Mechanical Properties

Figure 12 features a dotted line representing the experimental longitudinal compressive stress-strain curve. Inadequate pre-compression results in a gradual increase phase before the curve’s linear growth. After reaching its peak, the stress-strain curve for the 15° specimen drops sharply, indicating brittleness, while the curve for the 30° specimen declines gradually, exhibiting nonlinear characteristics. The nonlinear behavior observed in the 30° specimens is attributed to local delamination. The solid line depicts the longitudinal compressive stress-strain curve derived from finite element analysis. Figure 13a compares the overall strength and elastic modulus as determined by both finite element analysis and experimental methods. A quantifiable discrepancy exists between the finite element predictions and experimental outcomes, detailed in Figure 13b. The difference between finite element predictions and experimental findings is maintained within a 21% margin of error.

Figure 12a and Figure 12d display the stress-strain curves for 15° and 30° specimens with Type 1 prefabricated holes, respectively. The FEM analysis reveals that the 12 mm prefabricated hole specimens have lower strength compared to the 24 mm ones. The damage map (Figure 9) indicates that longitudinal compression damage tends to spread more between the two holes of the 12 mm prefabricated hole. Experimental results show some deviations from the finite element predictions. However, considering various factors, the experimental findings generally conform to this pattern. For instance, the stress-strain curve for specimen 15-1-24-(2) (Figure 12a) shows a brief plateau at 1% strain. Such an anomaly could be attributed to material cracks. The spacing between Type 1 prefabricated holes slightly affects the elastic modulus of 30° specimens but significantly impacts that of 15° specimens. The 24 mm prefabricated holes notably decrease the elastic modulus in 15° specimens. This demonstrates that the elastic modulus of braided composites with narrow braiding angles is highly sensitive to the horizontal spacing between two holes, with larger spacings increasing the material’s elastic modulus.

Figure 12b and Figure 12e display the stress-strain curves for 15° and 30° specimens with Type 2 prefabricated holes, respectively. The results indicate that the strength of the 12mm prefabricated hole specimens is marginally higher than that of the 24 mm ones. Additionally, the spacing between holes has minimal impact on the elastic modulus of woven materials with Type 1 prefabricated holes. The damage map (Figure 10) reveals significant damage expansion at both prefabricated holes when the spacing is 12 mm, indicating their joint participation in absorbing compression energy, thereby enhancing compression performance. Conversely, at a 24 mm spacing, significant damage occurs at only one prefabricated hole, suggesting its sole involvement in energy absorption. This isolation leads to material failure upon the hole’s destruction, resulting in weaker compression performance.

Figure 12c and Figure 12f illustrate the stress-strain curves for 15° and 30° specimens with Type 3 pre-drilled holes, respectively. Both FEM and experimental results indicate that the spacing between holes minimally affects the strength and elastic modulus of 15° specimens, as well as the strength of 30° specimens. However, for 30° specimens with Type 3 prefabricated holes, the hole spacing significantly impacts the elastic modulus. Larger spacings result in higher elastic moduli of the material. This demonstrates varying sensitivities of the elastic modulus in 15° and 30° specimens to different prefabricated hole configurations. The horizontal spacing between two holes influences the elastic modulus of 15° specimens. Spacing between two holes aligned with the knitting direction impacts the elastic modulus of 30° specimens. The spacing of other prefabricated hole types has minimal impact on the specimen’s elastic modulus.

According to Figure 13a, the strength hierarchy for specimens with prefabricated holes is as follows: specimens with longitudinal Type 2 holes exhibit the highest strength, followed by Type 3, and then Type 1 holes. Notably, the strength of the specimen 30-2-24 falls below anticipated levels. As depicted in Figure 12c, there is a sudden drop at the peak strength in the experiment, deviating from the expected nonlinear characteristics of the 30° specimens. This anomaly may result from internal defects within the specimen. Despite this, the FEM results remain within the permitted error margin. Regarding the elastic modulus, the hierarchy is: specimens with longitudinal Type 1 prefabricated holes have the highest modulus, followed by Type 2, and then Type 3 holes. Notably, the elastic modulus values for specimens 15-1-24 and 30-3-24 are exceptionally low, warranting further investigation. The reason for this phenomenon is unknown for the time being.

Figure 14 displays the strength loss for each specimen with prefabricated holes. Specimens with smaller braiding angles exhibit significantly higher strength loss than those with larger angles. The 15° specimens have an average strength loss of 38.21%, whereas the 30° specimens experience an average loss of 8.1%. The study of failure modes suggests that these differences are tied to the materials’ load-bearing mechanisms. For 15° specimens, the yarn’s direct bearing of longitudinal compressive loads means that openings significantly impact material strength. Generally, an increase in openings leads to a greater strength loss in 15° specimens. The 30° specimens primarily rely on the interaction between internal structures, making them less susceptible to strength loss from openings compared to 15° specimens.

## 5. Conclusions

This study demonstrates how various types of prefabricated holes affect uniaxial compression damage and failure in 15° and 30° specimens, assessing their porous damage tolerance ability.

Under longitudinal compression, the main load-bearing structure of the 15° specimen is the braided structure of yarns, and the damage develops from the longitudinal tensile crack on the surface to the transverse brittle fracture of the material. The main bearing mode of the 30° specimen is the interaction of internal structure, and the damage develops from the internal delamination of yarns and matrix to the transverse distribution of material. Different bearing modes lead to different mechanical responses of materials to prefabricated holes and different mechanical properties for different types of prefabricated holes.

Regarding porous defect tolerance, materials with larger braiding angles outperform those with smaller angles. The average strength loss is 38.21% for 15° specimens and 8.1% for 30° specimens. The analysis of failure mechanisms shows that the porous damage tolerance ability of braided composites is closely related to the bearing mode of material structure.

The discrepancy between the predicted and experimental elastic modulus and strength for specimens with two braiding angles is within 21%, showing that it is feasible to predict braided composites with prefabricated holes by using general FEM. However, the embedding constraints used in the large-size, high-volume model prevent the simulation of local delamination in 30° specimens using FEM. Only cell deletion indicates material failure, precluding the simulation of material nonlinearity.

## Figures and Tables

**Figure 1 materials-17-03821-f001:**
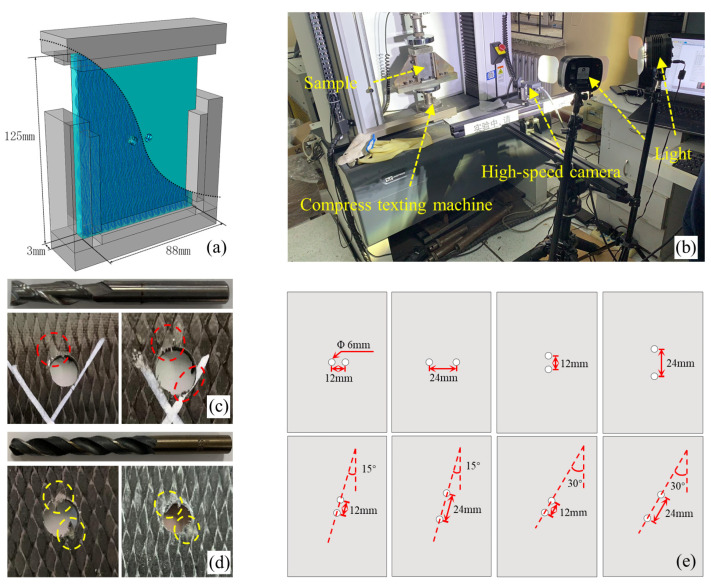
Compression test and specimens with prefabricated holes. (**a**) fixture of compression test and specimen sizes; (**b**) compression experiment and DIC equipment; (**c**) perforation performances of double-edge keyway milling cutters (The red circle shows the serious perforation damage.); (**d**) perforation performances of twist bits (The yellow circle shows the slight perforation damage.); and (**e**) 6 types of prefabricated holes.

**Figure 2 materials-17-03821-f002:**
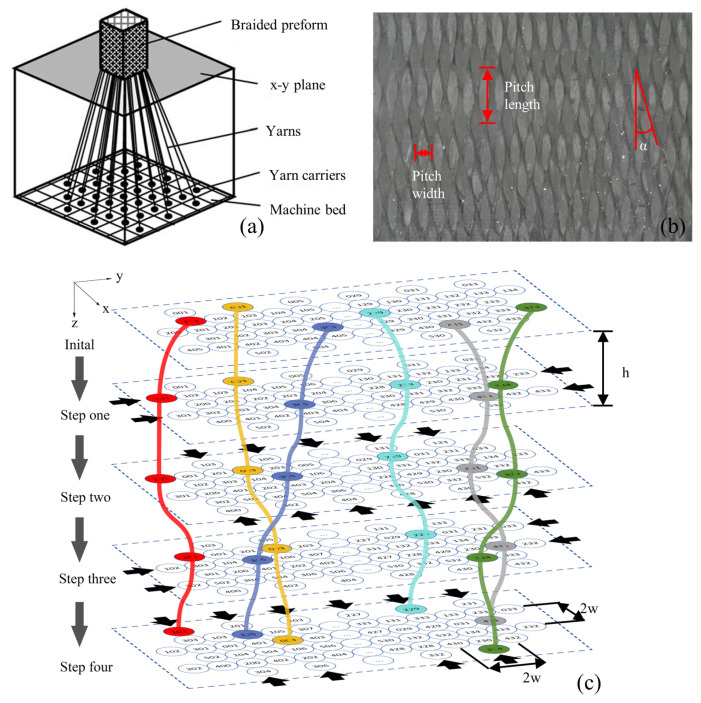
Four-step braiding principle. (**a**) 3D braiding process, (**b**) 3D braiding surface, and (**c**) schematic diagram of the 3D braiding process in one cycle.

**Figure 3 materials-17-03821-f003:**
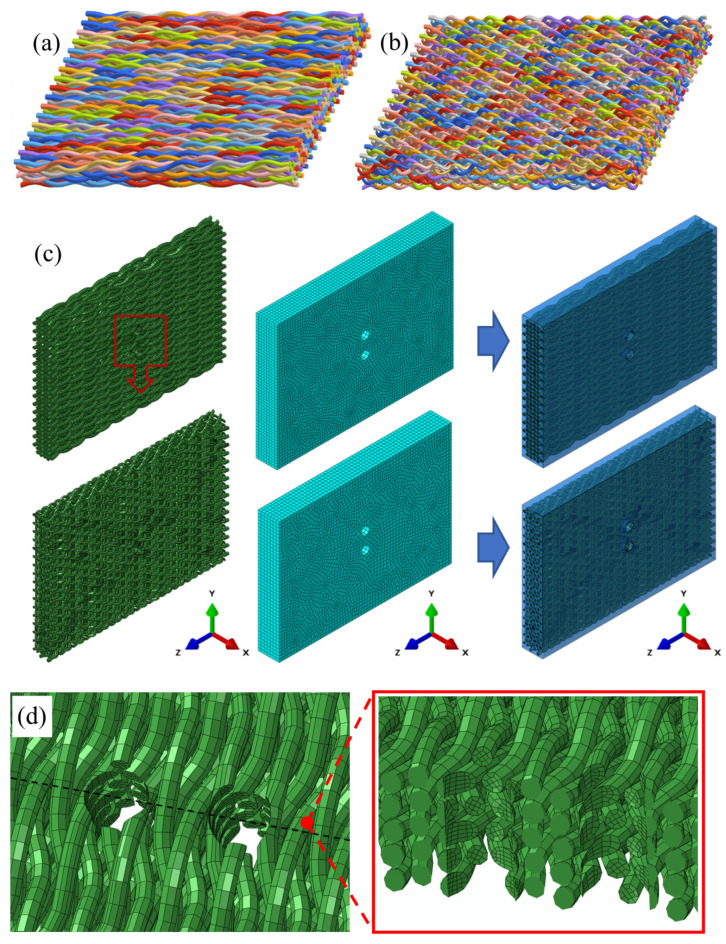
Model assemblies and meshes. (**a**) Braided structure of 15° specimens; (**b**) Braided structure of 30° specimens; (**c**) Model meshes and assemblies; and (**d**) The mesh of prefabricated holes.

**Figure 4 materials-17-03821-f004:**
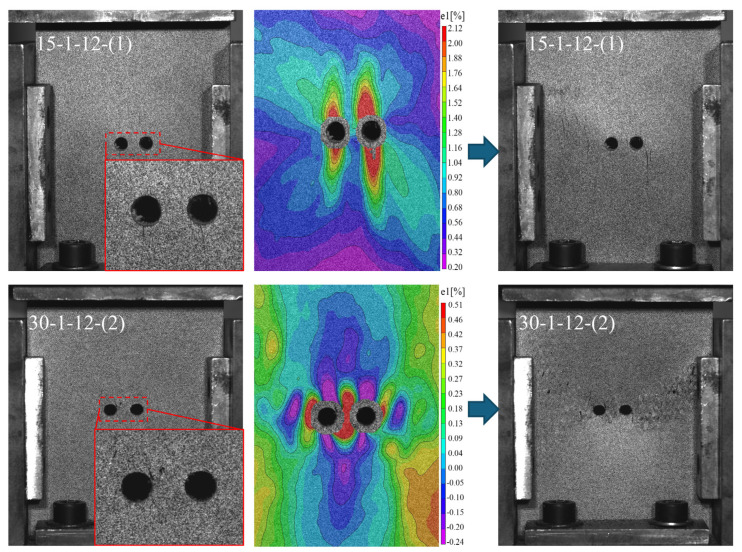
Damage extension patterns in braided composites with prefabricated holes at 15° and 30°.

**Figure 5 materials-17-03821-f005:**
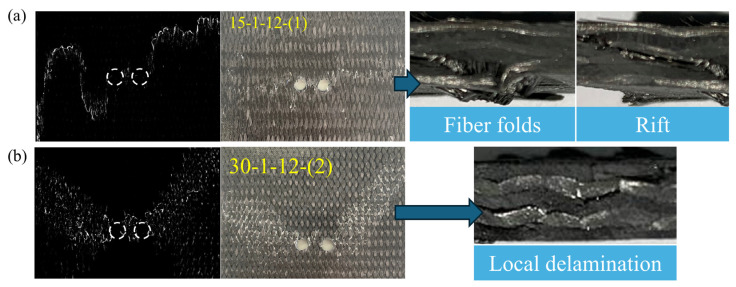
Failure modes of specimens with two different braiding angles. (**a**) 15° braided material (Example: 15-1-12-(1)). (**b**) 30° braided material (Example: 30-1-12-(2)).

**Figure 6 materials-17-03821-f006:**
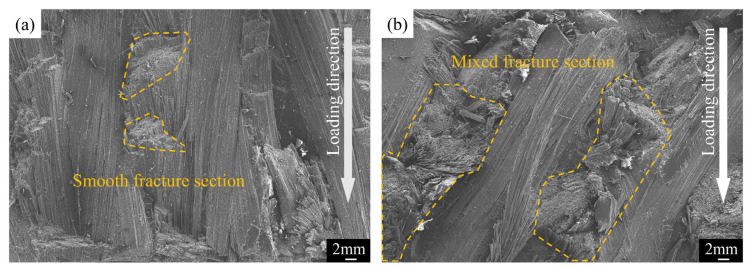
Fiber fracture sections in specimens with two different braiding angles. (**a**) specimen with 15 braided angles and (**b**) specimen with 30 braided angles.

**Figure 7 materials-17-03821-f007:**
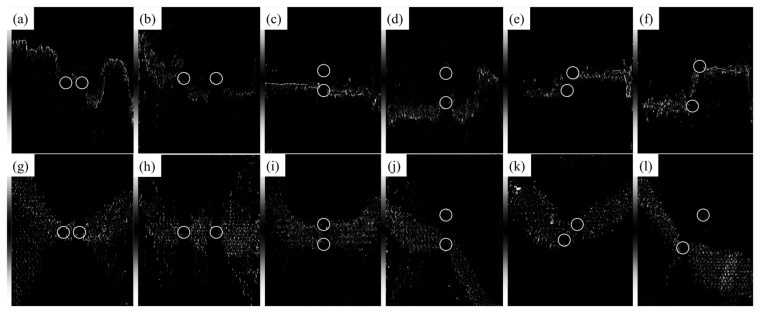
Internal damage after compression failure. (The circle is the identification of the hole position). (**a**) Specimen 15-1-12-(1), (**b**) Specimen 15-1-24-(2), (**c**) Specimen 15-2-12-(1), (**d**) Specimen 15-2-24-(2), (**e**) Specimen 15-3-12-(3), (**f**) Specimen 15-3-24-(1), (**g**) Specimen 30-1-12-(2), (**h**) Specimen 30-1-24-(2), (**i**) Specimen 30-2-12-(2), (**j**) Specimen 30-2-24-(1), (**k**) Specimen 30-3-12-(3) and (**l**) Specimen 30-3-24-(1).

**Figure 8 materials-17-03821-f008:**
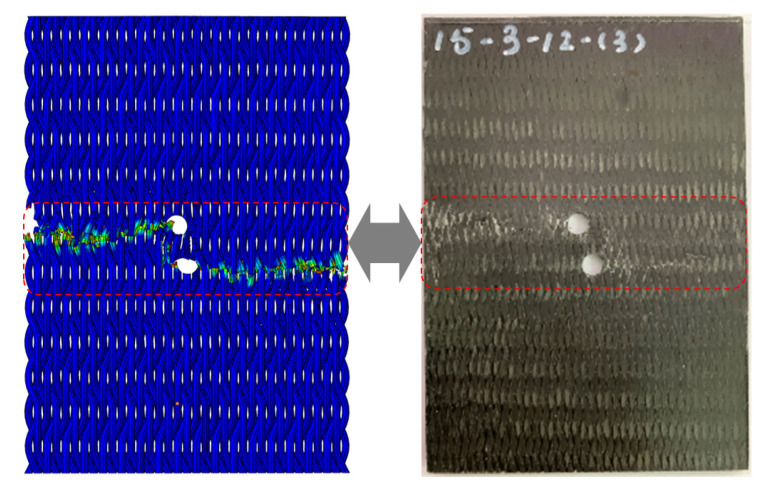
Failure comparison between finite element method and experiment (specimen 15-3-12).

**Figure 9 materials-17-03821-f009:**
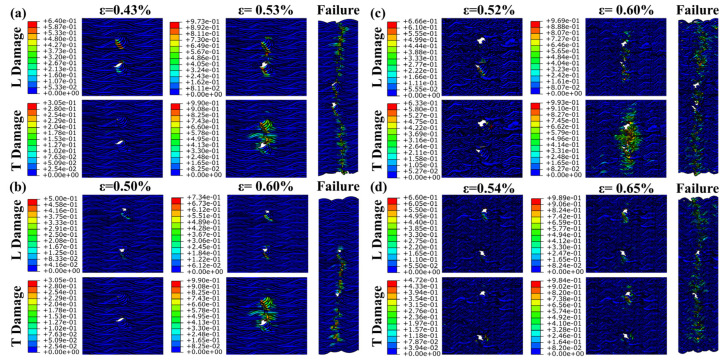
Damage extension and complete failure of the specimen with type 1 prefabricated hole under longitudinal compression. (**a**) the result of specimen 15-1-12, (**b**) the result of specimen 15-1-24, (**c**) the result of specimen 30-1-12, and (**d**) the result of specimen 30-1-24.

**Figure 10 materials-17-03821-f010:**
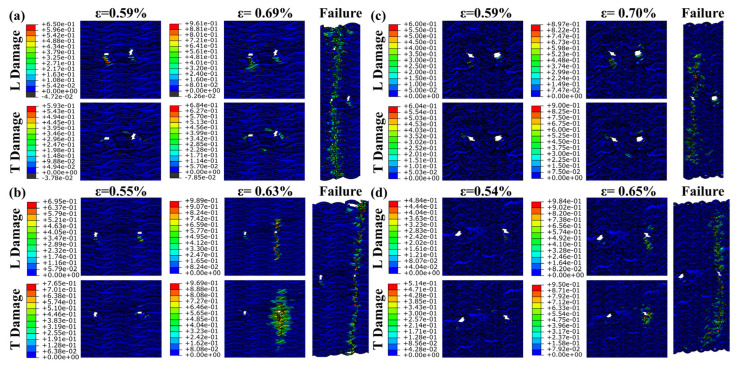
Damage extension and complete failure of the specimen with type 2 prefabricated hole under longitudinal compression. (**a**) the result of specimen 15-2-12, (**b**) the result of specimen 15-2-24, (**c**) the result of specimen 30-2-12, and (**d**) the result of specimen 30-2-24.

**Figure 11 materials-17-03821-f011:**
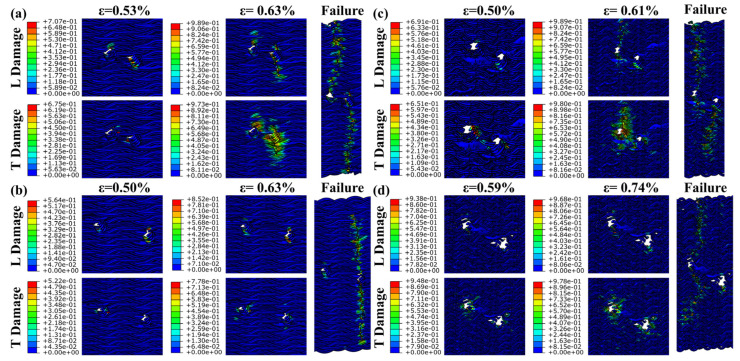
Damage extension and complete failure of the specimen with type 3 prefabricated hole under longitudinal compression. (**a**) the result of specimen 15-3-12, (**b**) the result of specimen 15-3-24, (**c**) the result of specimen 30-3-12, and (**d**) the result of specimen 30-3-24.

**Figure 12 materials-17-03821-f012:**
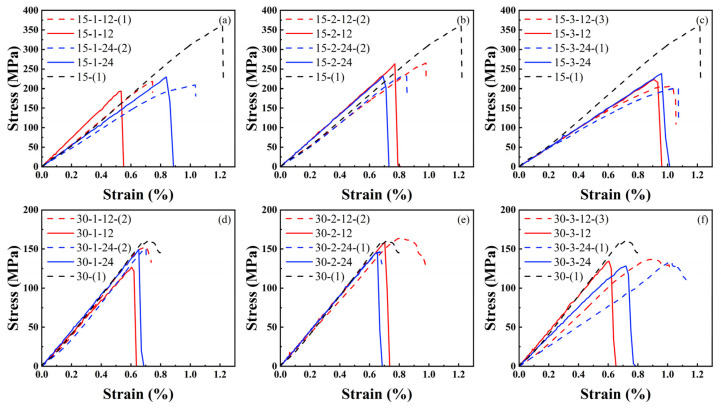
Longitudinal compression stress-strain curves of composites. (**a**) type 1 prefabricated holes of 15°, (**b**) type 2 prefabricated holes of 15°, (**c**) type 3 prefabricated holes of 15°, (**d**) type 1 prefabricated holes of 30°, (**e**) type 2 prefabricated holes of 30°, and (**f**) type 3 prefabricated holes of 30°.

**Figure 13 materials-17-03821-f013:**
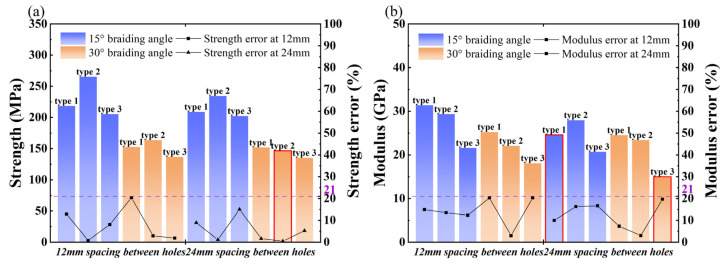
The modulus, strength, and error of 3D4d braided composites with different prefabricated holes at longitudinal compressions. (**a**) Material strength and prediction error, and (**b**) Material modulus and prediction error.

**Figure 14 materials-17-03821-f014:**
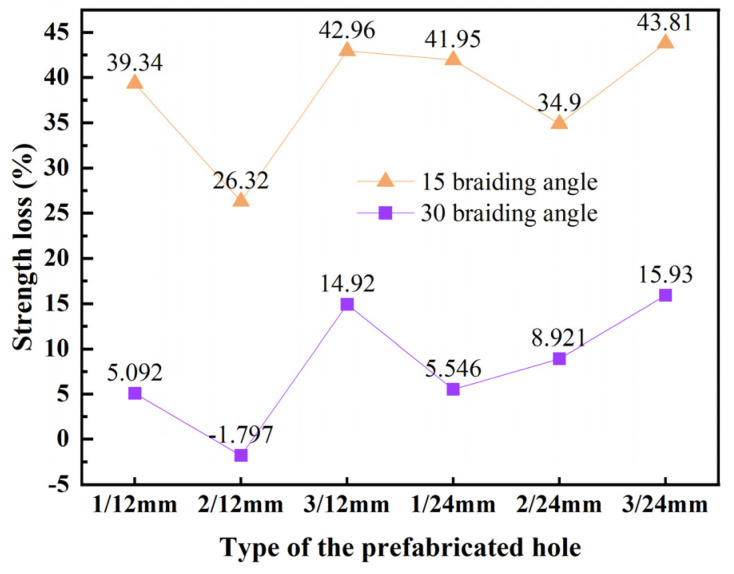
Strength loss of 3D braided composites with prefabricated holes compared with 3D braided composites without holes.

**Table 1 materials-17-03821-t001:** Main parameters of test materials.

Tape	Properties
Carbon fiber	TORAY T700-SC-12000-50B
Matrix	TED-86·epoxy resin matrix
Fabric structure	Three-Dimensional Four-Directional
Braiding Angles	15° and 30°
Measure	88 mm × 3 mm × 125 mm
Pitch length	41.0 + 1.0 mm (15°) 21.5 + 1.0 mm (30°)
Pitch width	11.0 + 0.5 mm (15°) 12.5 + 0.5 mm (30°)
Overall fiber volume content	(62.02 ± 1)% (15°) (63.98 ± 1)% (30°)
Fiber filling factor	75%

**Table 2 materials-17-03821-t002:** Geometric model parameters.

*α* (°)	*λ* (g/m)	*ρ* (g/cm^3^)	w (mm)	*h* (mm)	V_f_ (%)
15	0.8	1.8	0.9465	1.7662	60
30	0.8	1.8	1.14	0.99	50

**Table 3 materials-17-03821-t003:** Properties of constituents of 3D4d braided composites [12].

12K-T700 Carbon Fibers		TDE-86	
Longitudinal tensile elastic modulus, E1	230 GPa	Elastic modulus, E1 = E2 = E3	2.5 GPa
Longitudinal compressive elastic modulus, E1	130 GPa	Poisson’s ratio, μ12 = μ13 = μ23	0.35
Transverse elastic modulus, E2 = E3	15 GPa	Shear modulus, G12 = G13 = G23	1.28 GPa
Longitudinal Poisson’s ratio, μ12 = μ13	0.28	Density, ρ	1.19 g/cm^3^
Transverse Poisson’s ratio, μ23	0.4		
Longitudinal shear modulus, G12 = G13	20 GPa		
Transverse shear modulus, G23	5.36 GPa		
Longitudinal tensile strength, F1T	4900 MPa		
Density, ρ	1.8 g/cm^3^		

**Table 4 materials-17-03821-t004:** Fracture energy density of the carbon fiber and resin matrix [12].

GLt (N/mm)	GLc (N/mm)	GTt (N/mm)	GTc (N/mm)	Gmt (N/mm)	Gmc (N/mm)
8	8	1.5	1.5	1	1

## Data Availability

The raw/processed data required to reproduce these findings cannot be shared at this time due to legal or ethical reasons.

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
