# Peer review of "Compressive Failure Characteristics of 3D Four-Directional Braided Composites with Prefabricated Holes"

_materials, 2024, doi:10.3390/ma17153821_

Round 1

Reviewer 1 Report

Comments and Suggestions for Authors

The article is very interesting, the introduction chapter is well written, and the methodology and the description of the results are understandable in principle, but I still feel unsatisfied about the explanation of the holes, apparently a lot of space was devoted to them - but I cannot form an opinion about them after reading the article. I understand the search for a way to improve properties, but how does it quite work? 

With the automatic numbering of literature references, there were errors displayed in the text.

Abstract is a bit too general, please devote more to your successes in your work.

What do you mean by arguments in table 1, completely misses the word.

The conclusions are perhaps a little too elaborate.

kind regards

Reviewer 2 Report

Comments and Suggestions for Authors

Good article, well written, appropriate theoretical and experimental approach, well explained results, well justified conclusions. Worth to be published.

Please check and fix all the references throughout the manuscript

Reviewer 3 Report

Comments and Suggestions for Authors

Recommandation for the autours :

·         Mentioned figures are not appropriately mentioned in the text.

·         Figures 5, 6, 9, 10, 11, 12, and 13 should be improved.

·         In the abstract and introduction, you must announce clearly your findings

·         The drilling process will introduce initial damage, how do take it into account for your predictions? What precaution should be taken in experiment measurement and numerical one.

·         Specify the motivation of the chosen model. Clarify your objectives.

·         Compare your results to literature.

Round 2

Reviewer 3 Report

Comments and Suggestions for Authors

No further comments